# On the importance to consider the cloud dependence in parameterizing the albedo of snow on sea ice

Lara Foth[1], Wolfgang Dorn[1], Annette Rinke[1], Evelyn Jäkel[2], and Hannah Niehaus[3]

[1]Alfred Wegener Institute, Helmholtz Centre for Polar and Marine Research, Potsdam, Germany
[2]Leipzig Institute for Meteorology, University of Leipzig, Leipzig, Germany
[3]Institute of Environmental Physics, University of Bremen, Bremen, Germany

**Correspondence:** Wolfgang Dorn (wolfgang.dorn@awi.de)

**Abstract.** The impact of a slightly modified broadband snow surface albedo parameterization, which explicitly considers the cloud dependence of the snow albedo, is evaluated in simulations of a coupled regional climate model of the Arctic. The cloud dependence of the snow albedo leads to a more realistic simulation of the variability of the surface albedo during the snow melt period in late May and June. In particular, the reproduction of lower albedo values under cloud-free/broken-cloud conditions during the snow melt period represents an improvement and results in an earlier disappearance of the snow cover and an earlier onset of sea-ice melt. In this way, the consideration of the cloud dependence of the snow albedo results in an amplification of the two-stage snow-albedo/ice-albedo feedback in the model. This finds expression in considerably increased sea-ice melt during the summer months and ends up in a new quasi-stationary equilibrium in sea ice with statistically significantly lower sea-ice volume and statistically significantly lower summer sea-ice area.

## 1 Introduction

The surface albedo feedback effect is one of the main contributors to Arctic amplification (e.g., Pithan and Mauritsen, 2014; Hahn et al., 2021). It is known that "cloud cover normally causes an increase in spectrally integrated snow albedo" (Warren, 1982). Sensitivity studies already showed that "the increase in surface albedo with cloud cover can cause a doubling of the ice thickness" in model simulations (Shine and Henderson-Sellers, 1985). A couple of modeling studies addressed the importance of improving the sea-ice albedo parameterization (e.g., Liu et al., 2007; Toyoda et al., 2020); however, the cloud dependence of the surface albedo has not been considered explicitly in corresponding climate simulations by using appropriate parameterizations.

The cloud dependence of the surface albedo was demonstrated in several studies (Grenfell and Perovich, 2008; Gardner and Sharp, 2010; Stapf et al., 2020). It is caused by the different spectral characteristics of the incident radiation in a cloudy atmosphere (spectrally almost neutral) compared to cloudless conditions (strong spectral slope) and multiple surface–cloud interactions over highly reflecting surfaces. In the presence of clouds, the transmitted downward irradiance is weighted to shorter (visible) wavelengths, causing an increase in shortwave surface albedo, in particular in the polar regions where the solar zenith angle is large. This process seems to outweigh the albedo decreasing effect of a shift from mainly direct to rather diffuse irradiance in cloudy conditions, which decreases the surface albedo.

Even though climate models commonly calculate the atmospheric radiative transfer for clear sky and cloudy sky over a predetermined, but limited number of spectral bands, only few models use separate surface albedos for each of these spectral bands (e.g., van Dalum et al., 2020). Instead, most models use a broadband surface albedo, whereby spectral surface albedo variations are left out of consideration; and only few coupled climate models consider the cloud dependence in their broadband surface albedo parameterization (e.g., Boucher et al., 2020; Döscher et al., 2022).

The sea-ice surface can consist of dry snow, melting snow, bare and melting ice, melting and refreezing ponded ice, and sediment-laden ice (Light et al., 2022). The broadband albedo of these sea-ice surface subtypes is not only different but also differently influenced by clouds due to differences in the spectral albedo for visible and near-infrared wavelengths.

The present study focuses specifically on the parameterization of the snow albedo in the coupled Arctic regional climate model HIRHAM–NAOSIM (Dorn et al., 2019). The model serves as a testbed for analyzing the cloud effect on the albedo and

its potential consequences in fully coupled model simulations. HIRHAM–NAOSIM applies a broadband snow albedo parameterization, which has recently been supplemented by a simple cloud-cover dependence as suggested by Jäkel et al. (2019) on the basis of broadband surface albedo measurements carried out north of Svalbard in May/June 2017. A first evaluation of this cloud-cover-dependent snow albedo parameterization (hereinafter simply referred to as revised snow albedo parameterization) was already carried out by Jäkel et al. (2019) in a so-called offline application in which the parametrization was evaluated by

calculating the albedo with in-situ observed snow surface temperature and cloud cover. A more comprehensive evaluation of the complete albedo parameterization in offline and online application has recently been carried out by Jäkel et al. (2024) using data sets from different years and seasons.

In the present study, the performance of the revised snow albedo parameterization is evaluated only in online application, meaning in fully coupled model simulations, with focus on the role of the cloud-cover dependence. One purpose is to demon-

strate that the consideration of a simple cloud-cover dependence in the broadband snow albedo parameterization is able to emulate the cloud effect on snow surface albedo reasonably. The main purpose, however, is to demonstrate that this rather minor modification of the snow albedo parameterization has a statistically significant impact on the sea ice in a coupled model system due to its influence on the positive surface albedo feedback.

The sea-ice albedo parameterization used in this study is described in Sect. 2, the configuration of the various model sim-

ulations in Sect. 3.1, and the observations used for comparison in Sect. 3.2. The performance of the revised snow albedo parameterization is compared with the original parameterization and evaluated against the ratio between upwelling and downwelling irradiance and cloud cover observations in Sect. 4.1. Afterwards, the impact of considering a cloud-cover-dependent snow albedo parameterization on the modeled sea-ice evolution is demonstrated in Sect. 4.2.

## 2 Sea-ice albedo parameterization

HIRHAM–NAOSIM's sea-ice albedo parameterization was already described in great detail by Dorn et al. (2009) and is based substantially on version 2 of the sea-ice albedo schemes suggested by Køltzow (2007). The sea-ice surface consists of three subtypes: snow covered ice with albedo $\alpha_s$, melt ponded ice with albedo $\alpha_m$, and bare ice with albedo $\alpha_b$. The overall sea-ice

**Table 1.** Albedo values for cold, dry snow ($\alpha_{\mathrm{dry}}$, for $T \leq T_{\mathrm{dry}}$) and warm, wet snow ($\alpha_{\mathrm{wet}}$, for $T = 0\,^{\circ}\mathrm{C}$) and temperature threshold ($T_{\mathrm{dry}}$) in the original and the revised snow albedo parameterization. The revised snow albedo parameterization distinguishes between overcast conditions (cloud cover $\geq 95\,\%$) and cloud-free/broken-cloud conditions (cloud cover $< 95\,\%$). For temperatures $T_{\mathrm{dry}} < T < 0\,^{\circ}\mathrm{C}$ a linear transition between $\alpha_{\mathrm{dry}}$ and $\alpha_{\mathrm{wet}}$ according to Eq. (2) is applied.

| parameterization | clouds | $\alpha_{\mathrm{dry}}$ | $\alpha_{\mathrm{wet}}$ | $T_{\mathrm{dry}}\ (^{\circ}\mathrm{C})$ |
|---|---|---|---|---|
| original | – | 0.84 | 0.77 | −0.01 |
| revised | overcast | 0.88 | 0.80 | −3.0 |
| revised | non-overcast | 0.79 | 0.66 | −2.5 |

albedo $\alpha_{\mathrm{i}}$, as used for calculating the upwelling surface irradiance of the sea-ice-covered part of a model grid cell, is given as the weighted average of the respective albedos of the three subtypes according to the formula

$$\alpha_{\mathrm{i}} = c_{\mathrm{s}}\,\alpha_{\mathrm{s}} + c_{\mathrm{m}}\,\alpha_{\mathrm{m}} + (1 - c_{\mathrm{s}} - c_{\mathrm{m}})\alpha_{\mathrm{b}}\,, \tag{1}$$

where $c_{\mathrm{s}}$ is the fraction of the sea-ice surface covered with snow and $c_{\mathrm{m}}$ the corresponding fraction covered with melt ponds. With respect to the parameterization of $c_{\mathrm{s}}$ and $c_{\mathrm{m}}$, it is referred to Dorn et al. (2009, their equations (35) to (37)).

The subtype albedos $\alpha_x \in \{\alpha_{\mathrm{s}}, \alpha_{\mathrm{m}}, \alpha_{\mathrm{b}}\}$ are parameterized as a function of the surface temperature $T_{\mathrm{srf}}$ according to

$$\alpha_x = \alpha_{\mathrm{wet}} + (\alpha_{\mathrm{dry}} - \alpha_{\mathrm{wet}}) \min\left(1, \max\left(0, \frac{T_{\mathrm{srf}} - T_{\mathrm{wet}}}{T_{\mathrm{dry}} - T_{\mathrm{wet}}}\right)\right)\,, \tag{2}$$

where $\alpha_{\mathrm{wet}}$ and $\alpha_{\mathrm{dry}}$ are minimum and maximum values representing wet (melting) and dry (refreezing) surface conditions, respectively. $T_{\mathrm{wet}} = 0\,^{\circ}\mathrm{C}$ is the melting temperature of frozen water, at which the surface is considered as wet, and $T_{\mathrm{dry}} < T_{\mathrm{wet}}$ is a temperature threshold below which the surface is considered as dry. Eq. (2) represents a linear transition from $\alpha_{\mathrm{dry}}$ at $T_{\mathrm{srf}} = T_{\mathrm{dry}}$ to $\alpha_{\mathrm{wet}}$ at $T_{\mathrm{srf}} = T_{\mathrm{wet}}$.

While the parameters $\alpha_{\mathrm{wet}} = 0.16$, $\alpha_{\mathrm{dry}} = 0.36$, and $T_{\mathrm{dry}} = -2\,^{\circ}\mathrm{C}$ for melt ponded ice and the parameters $\alpha_{\mathrm{wet}} = 0.51$, $\alpha_{\mathrm{dry}} = 0.57$, and $T_{\mathrm{dry}} = -0.01\,^{\circ}\mathrm{C}$ for bare ice remained unchanged, the parameters for snow covered ice were defined separately for overcast conditions (cloud cover $\geq 95\,\%$) and non-overcast conditions (cloud cover $< 95\,\%$) in the revised snow albedo parameterization. The cloud-cover-dependent snow albedo parameters were suggested by Jäkel et al. (2019) and are listed in Table 1 together with the parameters used in the original snow albedo parameterization.

## 3 Data

### 3.1 Model simulations

Three pairs of two simulations each were carried out with the coupled regional climate model HIRHAM–NAOSIM (Dorn et al., 2019). The two simulations of each pair solely differ in the snow albedo parameterization. One simulation used the original snow albedo parameterization by Køltzow (2007) (hereinafter referred to as HNold), while the other used the revised snow

albedo parameterization by Jäkel et al. (2019) (hereinafter referred to as HNnew). In every other respect (e.g., the initialization, see below), the setup of the two simulations of each pair is identical.

HIRHAM–NAOSIM is applied over a circum-Arctic domain using rotated latitude-longitude grids with horizontal resolution of $1/4°$ ($\sim 27$ km) in the atmosphere component HIRHAM and $1/12°$ ($\sim 9$ km) in the ocean–sea ice component NAOSIM. More detailed information on the model components and their coupling is given by Dorn et al. (2019), and information on the model's cloud parameterization is given by Klaus et al. (2016). Information on the current model version, particularly with regard to recently introduced parameterizations, is given by Aue et al. (2023) in their supplementary material.

In the present study, all simulations were driven by ERA5 reanalysis data (Hersbach et al., 2020) at HIRHAM's lateral boundaries as well as HIRHAM's lower and NAOSIM's upper boundaries, which lie outside the coupling domain (defined as the overlap area of the components' model domains). For NAOSIM's open lateral boundaries, ORAS5 reanalysis data (Zuo et al., 2019) were used. While HIRHAM was always initialized with the corresponding ERA5 fields, the three pairs differ in their initial conditions for NAOSIM. While the first pair (P1) was initialized with fields from January 1, 2019, 00 UTC of an earlier long-term simulation, the two other pairs (P2 and P3) were started from rest with temperature, salinity, ice thickness, and ice concentration fields from ORAS5 and zero snow thickness.

Two of the pairs (P1 and P2) were carried out for the period 2019–2020 with nudging by which HIRHAM's prognostic fields, consisting of surface air pressure, horizontal wind components, air temperature, specific humidity, cloud liquid water content, and cloud ice content, were nudged to the corresponding ERA5 fields with a vertically uniform nudging time scale of 16.67 h (which corresponds to a nudging of 1 % per time step). The nudging was applied in order to reproduce the observed synoptic and large-scale atmospheric conditions and to enable the comparsion with measurements.

Pair P2 differs from P1 in the initial conditions for ocean and sea ice and was designed to investigate whether the initial conditions have influence on the simulation results. The different initial sea-ice thickness fields of the pairs P1 and P2 are provided in the supplementary material (Fig. S1). The third pair (P3) was carried out for the period 1979–2021 without nudging. This pair is used for analyzing the long-term sea-ice changes due to the revised snow albedo parameterization.

## 3.2 Observations

For the evaluation of the two snow albedo parameterizations, observational data from the Multidisciplinary drifting Observatory for the Study of Arctic Climate (MOSAiC) expedition (Shupe et al., 2022) were used. For the calculation of the surface albedo, irradiance measurements from an attended radiation station at MOSAiC's Met City location in the "Central Observatory" (MetCity) and from two autonomous atmospheric surface flux stations (ASFS30, ASFS50) that were deployed at different locations across the MOSAiC network were used. The irradiance measurements were carried out with upward and downward facing secondary-standard pyranometers; at Met City these were aspirated Eppley PSPs while at the ASFS these were internally-aspirated Hukseflux SR30-D1 pyranometers. More detailed information on the measurements are given by Cox et al. (2023d). The calculation of the surface albedo from irradiance measurements at the atmospheric surface flux stations, particularly from ASFS30, was necessary, since the regular albedo measurements at Met City were suspended from 12 May

to 17 June 2020 due to the transition of the research vessel *Polarstern* (Light et al., 2022); during this period ASFS30 was installed nearby the Met City location.

In addition, the surface albedo derived with the Melt Pond Detection (MPD) algorithm (Zege et al., 2015; Istomina et al., 2015b, a) from optical satellite observations (OLCI) were used for comparison. The data are produced as daily averages and gridded to a polar stereographic grid at a resolution of 6.25 km using the spectral-to-broadband conversion method described by Pohl et al. (2020). It is important to note that these optical measurements are limited to cloud-free conditions.

For cloud cover fraction, data from the ShupeTurner cloud microphysics product (Shupe, 2022) were used.

For consistency with the three-hourly model output, equivalent three-hourly means were calculated from all measurement data used in this study, except the OLCI data. For the OLCI albedo product, daily values were calculated by averaging all data points that fall within a single model grid cell. To compare model and observation, always data from the nearest model grid cell to the atmospheric surface flux station ASFS30 were selected. Model data for times without observational data were not taken into account.

## 4 Results

### 4.1 Evaluation of the cloud-cover-dependent snow albedo parameterization

The evaluation of the cloud-cover-dependent snow albedo parameterization was carried out for the pairs P1 and P2. The effect of the cloud dependence of the snow albedo is broadly similar in P1 and P2. Therefore, we discuss here only figures from P1 and provide the corresponding figures from P2 as supplementary material (Figs. S2 and S3).

Since this study focuses on the snow surface albedo, the analysis is restricted to the period where the incident solar radiation is relevant and where the sea-ice surface is almost entirely ($> 98$ %) covered by snow in the two model simulations. This period starts in mid-April and ends on 24 June 2020. It is further subdivided into an early cold period (15 April–25 May), with temperatures almost exclusively below $T_{\mathrm{dry}}$, and a late warmer period (25 May–24 June), with temperatures at or near the freezing point, where the snow begins to melt.

During the cold period, the sea-ice surface albedo in HNold is almost entirely defined by the albedo of dry snow without any variation (Figs. 1a and 1c). In contrast, the measurements from all three sites show distinct variations of the daily mean albedo (Fig. 1a) and even a broad spectrum of surface albedo values on a three-hourly basis (Fig. 1b). Although the measured albedo variations might not solely be attributable to changes in cloud cover, but also to local changes in the surface characteristics at the measurement site, which a climate model can not capture, it is obvious that a constant albedo in models is far from reality.

Since HNnew shows albedo variations, even if they are less pronounced than in the measurements (Figs. 1a and 1c), the implementation of a cloud dependence may be considered as one step into the right direction. Nevertheless, the values chosen for the albedo of cold, dry snow in HNnew, particularly for overcast conditions, appear to be too high as compared to the measurements. Even when considering that the three individual measurements depend on the local conditions at the measurement site, which might not be representative of the model's grid-cell area, the dry-snow albedo in HNnew is mostly higher than in the measurements.

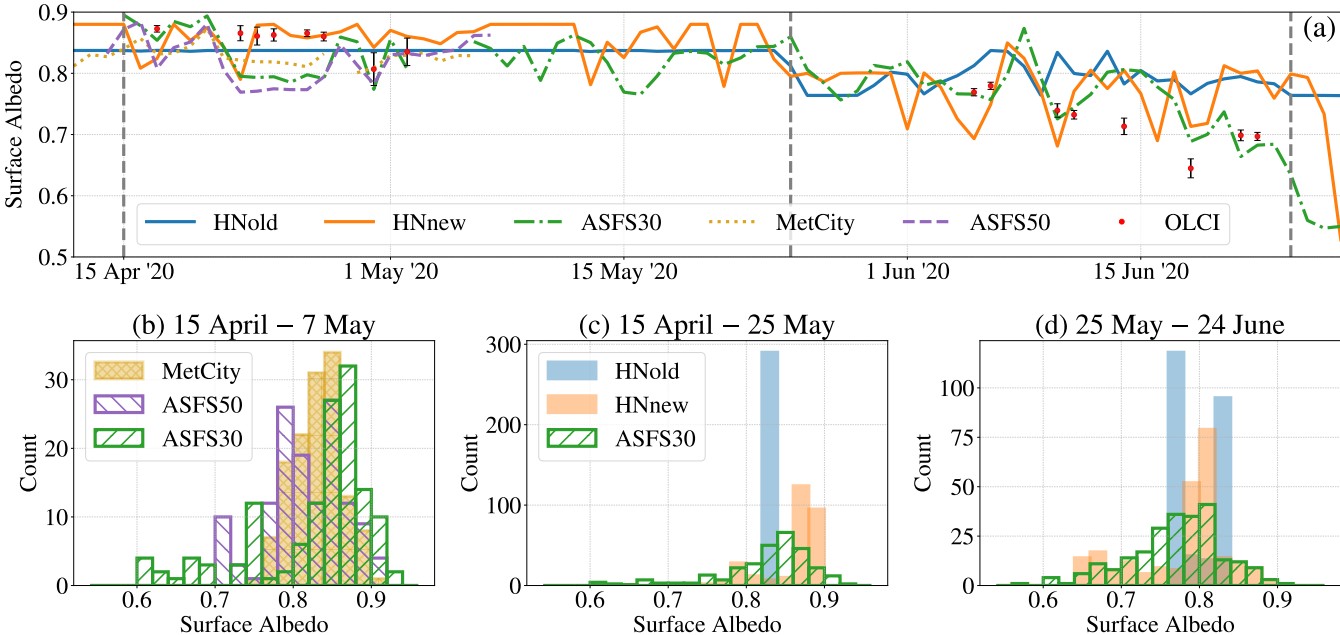

**Figure 1.** Top: (a) Time series of daily mean sea-ice surface albedo from the two P1 simulations (HNold and HNnew), from three irradiance measurements (ASFS30, MetCity, ASFS50), and from the satellite observations (OLCI) for the period from mid-April 2020 to end of June 2020. The dashed vertical lines indicate the beginning and the end, respectively, of the two evaluation periods, the cold period (15 April–25 May) and the snow melt period (25 May–24 June). Bottom: Frequency distribution of three-hourly mean sea-ice surface albedo from (b) the three irradiance measurements for the period where measurements from all three sites are available, (c) the two simulations and ASFS30 for the cold period, and (d) the two simulations and ASFS30 for the snow melt period.

In the cold period, overlapping OLCI albedo data are only available between mid-April and beginning of May, rather agreeing with the HNnew albedo than with the three pyranometer point measurements. However, the high albedo values in HNnew indicate overcast conditions, while the OLCI albedo always indicates cloud-free conditions. Interestingly, the variance of the satellite albedo increases (30 April and 2 May) where the point measurements become more similar. This might be caused by the spatial resolution closer to that of the model, not capturing the surface heterogeneity. In the snow melt period, the OLCI

albedo shows a progressive reduction with slight fluctuations that exceed the negative albedo trend of the two model runs. Nevertheless, the satellite albedo basically agrees with the point measurements, except for 14 June and 18 June where the satellite albedo is considerably lower than the albedo from the point measurements.

     During the snow melt period, HNold shows two distinct maxima of the sea-ice surface albedo, which relate to the albedo values of wet snow and dry snow (Fig. 1d). Values in between appear very seldom due to the small difference between $T_{\mathrm{dry}}$

and the freezing point. Albedo variations in HNold are solely a result of temperature fluctuations and do not reflect the large variations that appear in the measurements.

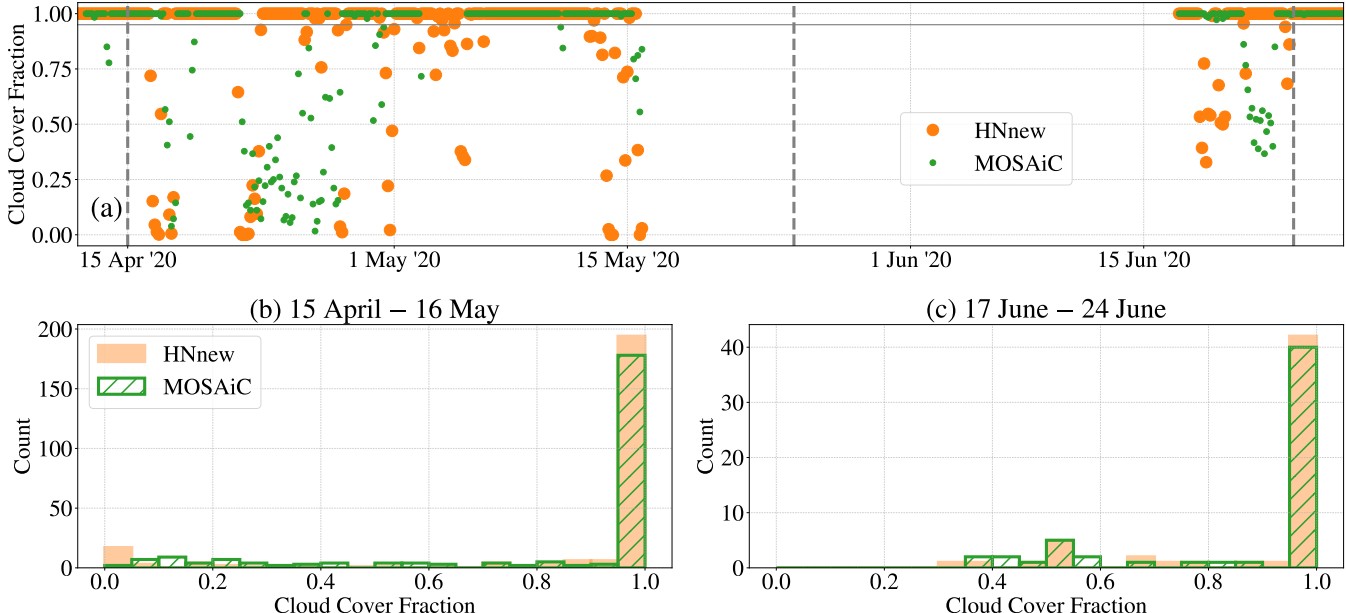

**Figure 2.** Top: (a) Time series of three-hourly mean total cloud cover fraction from the P1 simulation HNnew and the MOSAiC measurements for the period from mid-April 2020 to end of June 2020. The dashed vertical lines indicate the beginning and the end, respectively, of the two evaluation periods, the cold period (15 April–25 May) and the snow melt period (25 May–24 June). Bottom: Frequency distribution of three-hourly mean total cloud cover fraction from the simulation HNnew and the MOSAiC measurements for (b) the cold period and (c) the snow melt period. MOSAiC measurements are not available between 16 May and 17 June due to the transition of the research vessel *Polarstern*.

In comparison to HNold, the sea-ice surface albedo in HNnew shows a broad spectrum of albedo values in the range of 0.66 and 0.88 with a maximum between the two maxima of HNold. This spectrum is almost as broad as in the measurements, and also the mean values (0.78 vs. 0.77) and the standard deviations (0.06 each) statistically agree at the 95 % confidence level. In
particular, albedo values below 0.74, which can not occur in HNold, appear with similar frequency as in the measurements and indicate wet or partly wet snow under cloud-free/broken-cloud conditions. Since the amount of solar radiation that reaches the surface is generally larger under non-overcast than under overcast conditions, this feature promotes the melting of the snow cover.

In both periods, non-overcast conditions (here defined as total cloud cover fraction less than 95 %) appear in HNnew only
in less than one third of all cases (Figs. 2b and 2c), with a few cloud-free cases in the cold period and absolutely no cloud-free cases, but a couple of broken-cloud cases in the snow melt period. Although the simulation of clouds is generally regarded as one of the largest uncertainties in climate models (Flato et al., 2013), the cloud cover distribution in HNnew statistically agrees with the MOSAiC measurements in terms of mean value and standard deviation. Despite this statistical agreement, overcast and non-overcast conditions do not always appear at the same time (Fig. 2a). Consequently, the cloud-cover-dependent snow

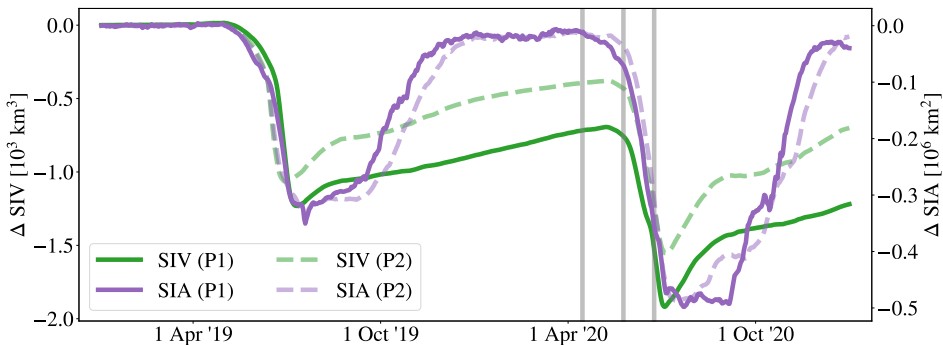

**Figure 3.** Time series of differences in daily mean Arctic sea-ice volume (SIV) and daily mean Arctic sea-ice area (SIA) between HNnew and HNold from P1 (solid lines) and P2 (dashed lines) for the entire simulation period 2019–2020. The gray vertical lines indicate the albedo evaluation periods described in the context of Fig. 1.

albedo can only statistically agree with the measurements. Nevertheless, the correspondence between the occurrence of cloud-free/broken-cloud conditions and the decrease in albedo is evident in both the HNnew simulation and the observations.

    Even though cloud-free/broken-cloud conditions appear not very often in HNnew, the intermittently occurring lower albedo values during the snow melt period amplify the melting of the snow. This is reflected in the fact that the residual atmospheric heat flux $Q_{\mathrm{res}}$ available for melting of snow and sea ice is on average 3.6 W m$^{-2}$ higher in HNnew between 25 May and

24 June, with the consequence that HNnew simulates an earlier disappearance of the snow cover of roughly one week compared to HNold (not shown here). Once the snow cover has disappeared, the surplus of solar radiation during the polar day is used to melt the sea ice, often by forming melt ponds on top of the ice, whereby the surface albedo decreases further. As a result, the ice melt starts not only earlier, but also the melt rate increases due to the decrease of the albedo during a time of high solar irradiance. Particularly in the short period from 25 June to 1 July, where the sea-ice surface mostly consists of bare ice and melt

ponds in HNnew and snow in HNold, the difference in $Q_{\mathrm{res}}$ between HNnew and HNold amounts on average to 45 W m$^{-2}$.

### 4.2   Impact on the modeled sea-ice evolution

The effect of this amplified two-stage snow-albedo/ice-albedo feedback is demonstrated in Fig. 3 by the differences between HNnew and HNold with respect to the evolution of the Arctic sea-ice volume and area. HNnew shows additional loss of sea-ice volume of around 1000 km$^3$ during the two periods of high solar irradiance. This additional loss of sea-ice volume is not fully

compensated during the subsequent freezing periods with the result that the sea-ice volume gradually decreases over the two years, presumably towards a new equilibrium with mostly thinner ice. The additional loss of sea-ice volume is larger in P1 than in P2, but the temporal evolution is qualitatively similar. This indicates that the specific initial conditions may have impact on the magnitude of the sea-ice loss, but not on the sea-ice loss in general.

    In combination with the additional loss of sea-ice volume, HNnew shows additional reduction of summer sea-ice area of

around 0.3 $\cdot 10^6$ km$^2$ during the first melting period and even around 0.5 $\cdot 10^6$ km$^2$ during the second melting period. In contrast

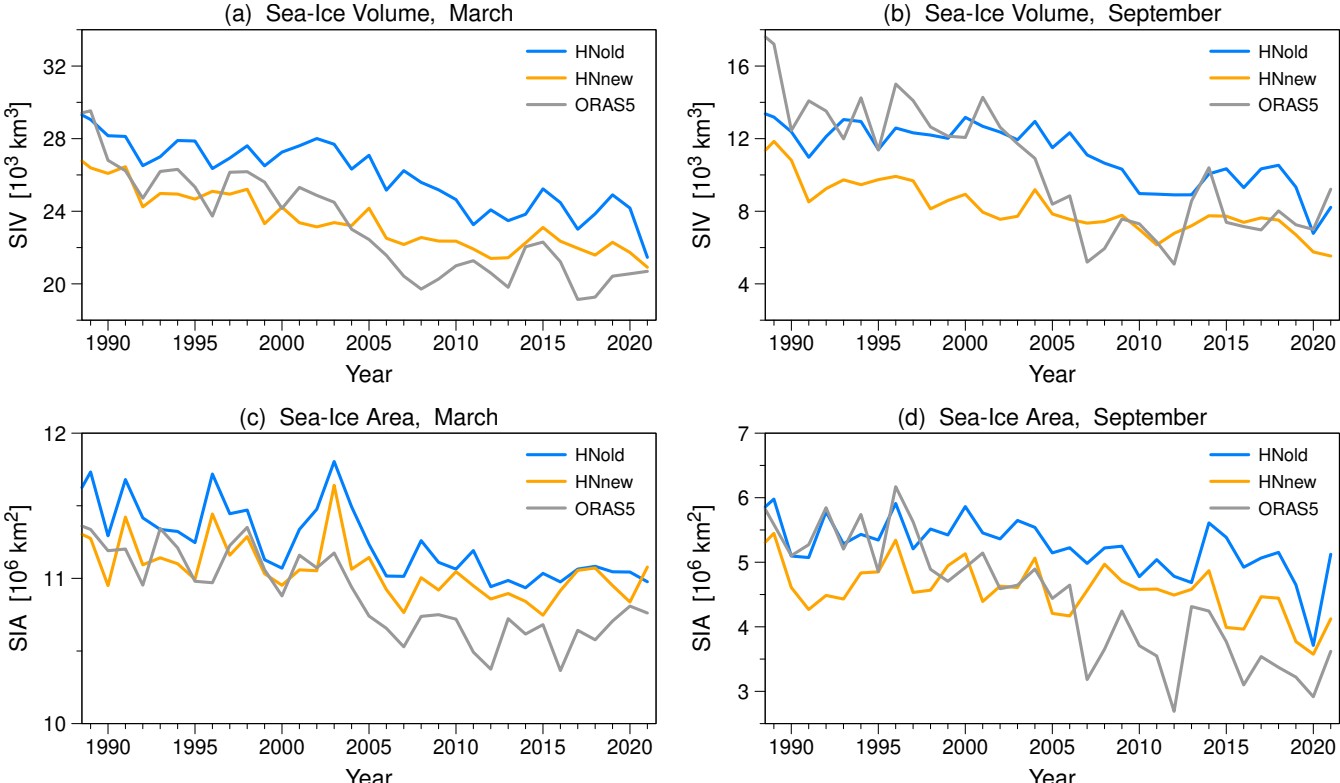

**Figure 4.** Temporal evolution of monthly mean Arctic sea-ice volume (a, b) and monthly mean Arctic sea-ice area (c, d) in March (a, c) and September (b, d) for the period 1989–2021 from the two P3 simulations (HNold and HNnew) and from ORAS5 reanalysis data. ORAS5 data were bi-linearly interpolated to the NAOSIM grid using distance-weighted averages for missing values.

to the sea-ice volume, the sea-ice area recovers almost completely during the cold season, but the sea-ice cover might become vulnerable for amplified reductions in subsequent melting periods due to the thinner ice. The effect on sea-ice area is similar in P1 and P2, with only slight differences in the magnitude of the reduction of summer sea-ice area.

The P3 simulations are used to demonstrate the long-term effect of the revised snow albedo parameterization on sea-ice volume and area. Dorn et al. (2007) showed that a coupled regional climate model needs about 6–10 years to arrive at a new quasi-stationary equilibrium in sea ice. They also demonstrated that this new equilibrium is independent from the initial ice conditions after 10 years. Therefore, the first 10 years of the P3 simulations are treated as spin-up time and neglected in the subsequent analysis.

Modeled sea-ice volume and area are here evaluated against corresponding data from the ORAS5 reanalysis (Zuo et al., 2019); and model biases consequently refer to ORAS5. It is known that ORAS5 sea-ice data slightly differ from satellite observations and other reanalysis products, as for instance PIOMAS (Zhang and Rothrock, 2003; Schweiger et al., 2011). Specific sea-ice biases in ORAS5 are discussed by Tietsche et al. (2018) and Zuo et al. (2019). Although the biases in HIRHAM–

NAOSIM are usually much larger than the differences of the various observational data products, the magnitude of the biases can differ when comparing with different data products. This particularly applies to the sea-ice volume, since sea-ice thickness observations are not assimilated into reanalysis systems. The PIOMAS reanalysis, for instance, exhibits a lower sea-ice volume than ORAS5 throughout the year, corresponding to a positive sea-ice thickness bias in ORAS5 (Tietsche et al., 2018) and over-/underprediction of thin/thick ice in PIOMAS (Schweiger et al., 2011). The HIRHAM–NAOSIM biases specified below refer only to ORAS5 as a benchmark and must be interpreted taking account of ORAS5 biases.

The modeled sea-ice volume in March (Fig. 4a) is in HNnew on average 2540 km$^3$ lower than in HNold. Compared to the sea-ice volume from ORAS5, the model's mean bias reduces from 2823 km$^3$ to 283 km$^3$. In September (Fig. 4b), the sea-ice volume in HNnew is on average 2929 km$^3$ lower than in HNold, but the difference to ORAS5 is only reduced from 2004 onwards. Averaged over the entire period, the sign of the difference to ORAS5 changes from positive in HNold to negative in HNnew and the magnitude of the bias in sea-ice volume even increases in HNnew. The strong decline of the September sea-ice volume from 2001 to 2007, as it appears in ORAS5 ($-9.1 \cdot 10^3$ km$^3$), is not reproduced either in HNold ($-1.6 \cdot 10^3$ km$^3$) or in HNnew ($-0.6 \cdot 10^3$ km$^3$). This shortcoming can obviously not be solved by improving the snow albedo parameterization. It likely requires improvements in other process descriptions.

In contrast to the sea-ice volume, the changes in sea-ice area are less pronounced. The mean bias in sea-ice area in March (Fig. 4c) is reduced in HNnew ($0.17 \cdot 10^6$ km$^2$) as compared to HNold ($0.36 \cdot 10^6$ km$^2$), but the difference between HNold and HNnew is less pronounced at the end of the simulation period than in the early years. In September (Fig. 4d), the magnitude of the changes in sea-ice area is on average larger than in March, and the difference to ORAS5 reduces from $0.83 \cdot 10^6$ km$^2$ in HNold to $0.15 \cdot 10^6$ km$^2$ in HNnew. The overall positive bias in September sea-ice area originates primarily from the overprediction of sea-ice area after 2006 and is likely related to the aforementioned strong decline of the observed sea-ice volume in the 2000s, which is not present in the simulations.

While the sea-ice volume in HNnew is lower than in HNold in every year, there are a few years at the end of the simulation period where the sea-ice area in HNnew and HNold is similar in magnitude, particularly in March. In contrast to the simulation pairs P1 and P2, the HNnew and HNold simulations of P3 were running without nudging to ERA5 with the result that the atmospheric circulation may diverge in the two simulations. This fact may implicate differences in the modeled sea-ice distribution that are not induced by differences in the model physics as exemplified by Dorn et al. (2012) for the years 1995 and 2007 in HIRHAM–NAOSIM ensemble simulations. For the same reason, changes in sea-ice volume and sea-ice area from one year to the next may be positive in one simulation and negative in the other. Consequently, year-to-year changes in sea-ice volume and sea-ice area can not be regarded as a direct consequence of the modified snow albedo parameterization, but as a manifestation of internally generated model variability.

Nonetheless, the average long-term effect of the revised snow albedo parameterization can be quantified from the mean difference in the seasonal cycle (Fig. 5). While HNold overpredicts the sea-ice volume throughout the year compared to ORAS5, HNnew shows a good agreement with ORAS5 from January to May (deviations of less than 2 %), but underpredicts the sea-ice volume from July to October even more than HNold overpredicts it. In contrast, both HNold and HNnew overpredict the sea-ice area in all months, except for HNnew in August, but HNnew is always closer to ORAS5. The annual mean bias in

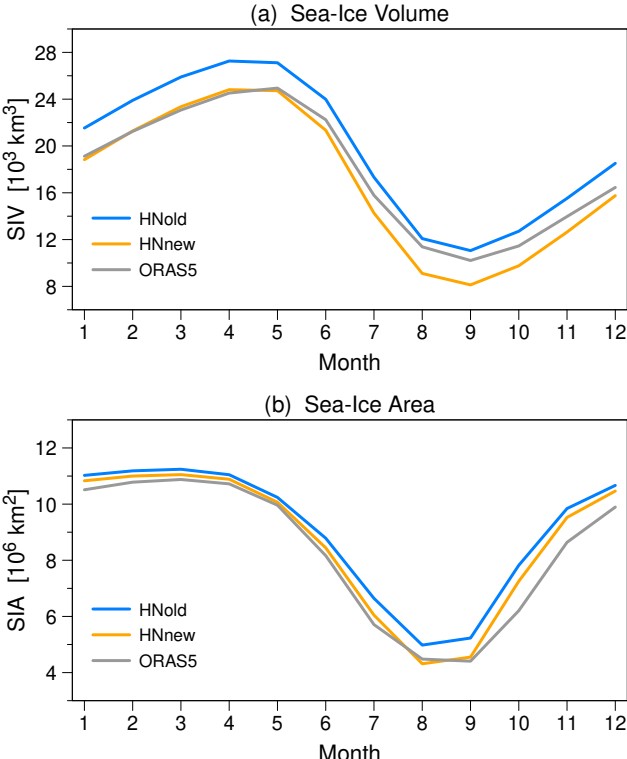

**Figure 5.** Mean seasonal cycle of Arctic sea-ice volume (a) and Arctic sea-ice area (b) for the period 1989–2021 from the two P3 simulations (HNold and HNnew) and from ORAS5 reanalysis data. ORAS5 data were bi-linearly interpolated to the NAOSIM grid using distance-weighted averages for missing values.

sea-ice area amounts to $0.70 \cdot 10^6$ km$^2$ in HNold and $0.34 \cdot 10^6$ km$^2$ in HNnew, while the annual mean bias in sea-ice volume amounts to 1874 km$^3$ in HNold and $-867$ km$^3$ in HNnew. This difference in sea-ice volume between HNnew and HNold

240 ($-2742$ km$^3$) corresponds to approximately 0.2 m (11 %) thinner sea ice in HNnew.

The differences between HNnew and HNold in both the sea-ice volume and the sea-ice area are statistically significant at the 99 % confidence level for all months (Figs. 6a and 6b). While there is only a weak seasonal cycle in the sea-ice volume difference that faintly indicates increased melting from May to July (Fig. 6a), the sea-ice area differences between HNnew and HNold show a distinct seasonal cycle with maximum during and subsequent to the melting period and minimum during the

245 cold season (Fig. 6b). This fact agrees with the earlier finding from the P1 and P2 simulations that the sea-ice area recovers almost completely during the cold season in spite of lower sea-ice volume.

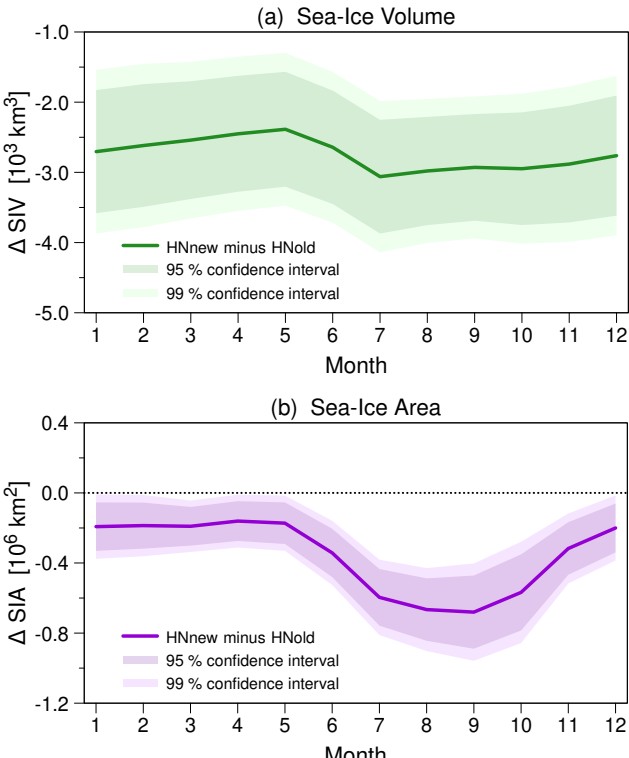

**Figure 6.** Seasonal cycle of the mean difference in Arctic sea-ice volume (a) and Arctic sea-ice area (b) between HNnew and HNold from P3 for the period 1989–2021. The shaded areas represent the 99 % confidence interval (in light color) and the 95 % confidence interval (in somewhat darker color), determined from monthly mean values according to Welch's $t$-test (Welch, 1938).

## 5  Summary and conclusions

The broadband snow albedo parameterization was supplemented by a cloud dependence to take account of the spectral shift of the transmitted downward irradiance towards shorter wavelengths in the presence of clouds. The implemented cloud dependence can be regarded as simplified approach, since it only distinguishes overcast conditions from non-overcast conditions. Given that overcast conditions appear more often than cloud-free/broken-cloud conditions, one could assume that the effect of the higher albedo for overcast conditions in HNnew compared to the cloud-independent albedo in HNold overcompensates the effect of the lower albedo for cloud-free/broken-cloud conditions. Actually, the mean albedo of the entire evaluation period from 15 April 2020 to 24 June 2020 is almost equal in HNnew and HNold, which basically results from higher albedo values during the cold period and lower ones during the snow melt period in HNnew.

As the higher albedo during the cold period can be considered as entirely irrelevant for melting of snow or sea ice, the albedo differences between model simulation and observation during this period do not play an important role in the overall performance of the revised snow albedo parameterization. There are also indications that the observed surface albedo variations

are not merely a result of the cloud cover, but rather dependent on the liquid water content of the clouds as shown by Stapf et al. (2020). Also changing surface characteristics might play a role, for instance due to snow metamorphosis or snow drift. However, it is challenging to find an adequate relation between the modeled liquid water path and the snow albedo that describes the cloud effect on the albedo in a realistic way, and it is nearly impossible to derive small-scale surface characteristics from the available variables of a climate model with spatial scales in the order of several kilometers.

In contrast to the cold period, the lower albedo during the warmer period, where the snow becomes wet and starts to melt, is important. The cloud-cover dependence in combination with the temperature dependence of the albedo appears to be sufficient in order to reproduce the observed albedo variations from a statistical point of view. In this view, the revised snow albedo parameterization represents an improvement compared to the cloud-independent original snow albedo parameterization. This also suggests that it is not absolutely necessary to implement a physically more reasonable, but also computationally more complex waveband-dependent albedo parameterization, which implicitly would include a cloud effect on the net solar irradiance at the snow surface.

The most important improvement is that the intermittently occurring lower albedo values under cloud-free/broken-cloud conditions lead to an increased heat supply at the snow surface that promotes the melting of snow. The implication in the coupled model systems is an earlier disappearance of the snow cover and an earlier onset of sea-ice melt, which translates into an amplification of the two-stage snow-albedo/ice-albedo feedback. This finds expression in additional loss of sea-ice volume of around 1000 $km^3$ and additional reduction of summer sea-ice area of around $0.3 \cdot 10^6 \, km^2$ during the first melting period. Over the next few years, the sea-ice changes gradually increase and result in a new quasi-stationary equilibrium in sea ice with statistically significantly lower sea-ice volume and statistically significantly lower summer sea-ice area beyond the 99 % confidence level. This new equilibrium is closer to reanalysis data for the most part, even though biases still exist. These biases are very likely a consequence of uncertainties in other parameterizations, as for instance in the treatment of lateral ice melt versus ice melt at the surface, as the different sign of the HNnew biases in summer sea-ice volume and area suggests.

The considerable impacts of the snow/ice albedo parameterization on the sea-ice simulation were already discussed in earlier studies (e.g., Liu et al., 2007; Dorn et al., 2007, 2009). These studies were based on inherently different representations of the sea-ice surface albedo, while the present study has focussed on only one aspect of the albedo parameterization, namely the cloud dependence of the snow albedo. Nevertheless, the latter has comparable impact on the sea-ice simulation and provides a simple opportunity to improve coupled model simulations as long as counteracting shortcomings in other model parameterizations are disregarded.

The ice-albedo feedback is of course also influenced by the albedo of both bare ice and melt ponds. Also the parameterization of the respective fractions of snow, bare ice, and melt ponds are a matter of importance for a realistic simulation of the snow-albedo/ice-albedo feedback as shown by Jäkel et al. (2024). All these aspects have been left aside in this study. Nevertheless, the statistically significant impact on the simulated sea-ice volume and area due to a rather minor modification of the snow albedo parameterization indicates how important it is to develop more realistic albedo parameterizations on the basis of observations, especially for coupled model systems.

*Code and data availability.* Because HIRHAM–NAOSIM contains source code being subject to intellectual property rights, the model source code is not freely accessible. Access to the model repository at https://gitlab.awi.de/wdorn/hirham-naosim will be granted to individuals on request. HIRHAM–NAOSIM data are available at the tape archive of the German Climate Computing Center (DKRZ) via persistent URLs (Dorn, 2024a, b, c). Measurements from Met City and from the Atmospheric Surface Flux Stations are available at the Arctic Data Center (Cox et al., 2023a, b, c). The ShupeTurner cloud microphysics product is available at the ARM Archive (Shupe et al., 2022). The OLCI raw data are available from https://ladsweb.modaps.eosdis.nasa.gov/archive/allData/450/, and the processed MPD albedo product is available from https://data.seaice.uni-bremen.de/databrowser/#p=MERIS_OLCI_albedo. ORAS5 reanalysis data are available from the Copernicus Climate Change Service, Climate Data Store (2021).

*Author contributions.* All authors contributed to conception and design of the study. WD performed the model simulations. HN processed the surface albedo from the satellite data. LF conducted all other data processing, produced Figures 1 to 3, and analyzed the results. WD produced Figures 4 to 6 and wrote the final draft of the manuscript with input from all co-authors. All authors contributed to manuscript revision, read, and approved the submitted version.

*Competing interests.* The authors declare that no competing interests are present.

*Acknowledgements.* WD, AR, EJ, and HN acknowledge the funding by the Deutsche Forschungsgemeinschaft (DFG, German Research Foundation) project 268020496 TRR 172, within the Transregional Collaborative Research Center "ArctiC Amplification: Climate Relevant Atmospheric and SurfaCe Processes, and Feedback Mechanisms (AC)[3]". WD and AR acknowledge the funding by the European Union's Horizon 2020 research and innovation framework programme under Grant agreement no. 101003590 (PolarRES project). This work used resources of the Deutsches Klimarechenzentrum (DKRZ) under project ID aa0049. Some of the data used in this manuscript were produced as part of the international Multidisciplinary drifting Observatory for the Study of Arctic Climate (MOSAiC) with the tag MOSAiC20192020 and the project ID AWI_PS122_00. Data from autonomous atmospheric surface flux stations were provided by the University of Colorado / NOAA surface flux team. Radiation data from Met City were collected by the Atmospheric Radiation Measurement (ARM) User Facility, a DOE Office of Science User Facility managed by the Biological and Environmental Research Program. We thank all those who contributed to MOSAiC and made this endeavor possible (Nixdorf et al., 2021). We also acknowledge support by the Open Access publication fund of the Alfred-Wegener-Institut, Helmholtz-Zentrum für Polar- und Meeresforschung. Finally, we thank Manfred Wendisch and Matthew Shupe for their helpful comments on an earlier draft of this paper.

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
