# Peer review of "On the importance to consider the cloud dependence in parameterizing the albedo of snow on sea ice"

_EGUsphere, 2023_

## Author Response (AR1)

**Reply to Referee #1**

We thank Referee #1 for the time and effort he/she spent reading the manuscript and providing extensive comments. We also appreciate the enormous number of potential suggestions for improvements and further investigations. In the following, a point-by-point response to all major and minor concerns (in italics) is given. Since the changes/additions made in the manuscript are considerable, we can not mention all of them in detail and refer the reader to the marked-up manuscript version provided together with this response.

**Major Concerns**

*1. Interannual Variability of the Pan-Arctic Findings*

*My largest concern is that the model-based results regarding sea ice extent and volume are derived from only one melt season (2019/20). Without presenting evidence that this season is somehow representative of the others, the model-based results represent a single number drawn from an unknown probability distribution. It is therefore unclear to me how to interpret the results with respect to the underlying feedback mechanisms. Without investigating other years, results concerning the role of the albedo parameterization cannot safely be interpreted outside of the specific study period.*

[REPLY] In the course of the manuscript revision, two long-term sensitivity simulations for the period 1979-2021 were carried out, one with the original snow albedo parameterization, the other with the revised snow albedo parameterization. These two simulations were running without nudging to ERA5 data, meaning that the model is allowed to diverge from the observed atmospheric circulation so that internal variability becomes more relevant. The results of the two simulations are discussed in the revised version of the manuscript and demonstrate that the additional loss of sea ice volume and extent appears independently from the specific conditions in 2019/20, even though year-to-year changes may be affected by internally generated model variability. The two long-term simulations confirm our assumption that the amplified surface albedo feedback due to the cloud-cover dependence of the snow albedo parameterization results in a new quasi-stationary equilibrium in sea ice with mostly thinner ice.

*2. Separability of Cloud and Temperature Dependence*

*My secondary concern is that the authors have implemented both a cloud dependence and a temperature dependence in their albedo parameterisation, but often report results (such as in the abstract, and title) as if they are simply evaluating the new cloud dependence. It is important and currently unclear to me how much of the results are due to the temperature dependence, and how much is due to the cloud-dependence. Regarding the title, we will only actually know the importance of the cloud dependence in this study's albedo parametrization once the temperature effect that the authors have included has been established and controlled for.*

[REPLY] The new snow albedo parameterization basically uses the same temperature dependence as the original snow albedo parameterization, only the values of the parameters are separately defined for overcast and non-overcast conditions. When averaging the albedo for overcast and non-overcast conditions relative to the frequency of the conditions, the albedo values are

almost equal in the two parameterizations. This means that changes in the temperature dependence effectively depend on the cloud dependence. There is no cloud-independent temperature dependence which one might evaluate without further sensitivity experiments. In the sensitivity experiments presented in our study, all differences between the two model simulations can be considered as a consequence of the additionally introduced cloud dependence. We have tried to make this point clearer in the revised version of the manuscript.

*3. How skillful is this model in the study period?*

*The impact of this study is tightly linked to the skill of the model in the study period, so it is striking that this is not discussed. I can only assume (see my minor comments below) that the model run began with the sea ice in the right place and with roughly the right thickness and overlying snow depth based on satellite observations. But after initialisation does the sea ice move, grow and melt in a realistic way? If the sea ice doesn't behave correctly in the model, then the real-world impact of getting the albedo right will not be available through the model study. The authors should present their findings in the context of the model's underlying ability to investigate these questions.*

*On this note, I was surprised that the authors didn't contextualise the additional loss of sea ice volume and extent with regard to the "truth" of the model. i.e. do these extra reductions move the modelled sea ice extent/volume closer or further away from the truth per satellites? That is to say, was it over or underestimating these variables before?*

[REPLY] The model used for this study is not a new development. The model has been used for a couple of Arctic climate studies in the past, and its skill to simulate sea ice has been evaluated by Dorn et al. (2019, https://doi.org/10.3390/atmos10080431) and Yu et al. (2020, https://doi.org/10.5194/tc-14-1727-2020). The model was recently also applied to the MOSAiC period (2019/20) by Aue et al. (2023, https://doi.org/10.3389/feart.2023.1112467). They found that the model is able to capture the synoptic situation and produce a realistic spatial pattern of sea-ice concentration changes. However, it is not the aim of the paper to demonstrate that the model is better than other models or that the sea-ice simulation is closer to reality by considering a cloud dependence in the albedo parameterization. Even though the latter is actually the case in our model simulations, we had decided to exclude praising our own model, since we aware that the grade of sea ice in a coupled model depends on many parameterizations, not only on the albedo parameterization. Some of these parameterizations are usually subject of a kind of tuning to obtain more realistic sea ice in the model as for instance discussed by Mauritsen et al. (2012, https://doi.org/10.1029/2012MS000154). Therefore, it doesn't mean much whether the modeled sea ice is better or worse than before as long as tuning of parameters in other process descriptions, which are not well established by observations, were applied to counteract the biases of a previously more unrealistic albedo parameterization. In particular, it doesn't disqualify the added value of considering the cloud dependence in parameterizing the snow albedo. In the revised version of the manuscript, we evaluate sea-ice volume and extent against corresponding data from the ORAS5 reanalysis (notwithstanding that ORAS5 might not be the "truth" as well) to show that the consideration of a cloud-dependent snow albedo parameterization can lead to reduced model biases in the sea-ice simulation. In addition, we point to specific model shortcomings which can not be solved by improving the snow albedo parameterization and require improvements in other process descriptions. For reasons of clarification, we also explicitly indicate in the introduction that the used model serves as a testbed for analyzing the cloud effect on the albedo and its potential consequences in fully coupled model simulations.

**Minor Concerns**

*Manuscript structure: There are some peculiarities with the manuscript structure: the paper as a whole would definitely benefit from deeper analysis and a dedicated discussion section. To illustrate the need for a discussion section, the first paragraph of Conclusions ( L140) is a really nice piece of analysis, but cannot reasonably be described as a conclusion. I think the lack of dedicated discussion has also led to the omission of the key methodological details and considerations that I've outlined above and below.*

[REPLY] The manuscript was initially designed as "brief communication" in The Cryosphere, but turned out being a little too long for this manuscript category. Due to the design of the manuscript as brief communication, the manuscript was concise and a lengthy discussion of the results had been omitted. Since a "brief communication" is not feasible anymore, we have extended the introduction, we have added further methodological details, particularly the new section "Sea-ice albedo parameterization", previously omitted for reasons of compactness, we have separated the "Results" section into two subsections, where the second subsection "Impact on the modeled sea-ice evolution" is almost completely new, we have replaced the "Conclusions" by "Summary and conclusions", where we now discuss the results in greater detail, and we have added further references where appropriate.

*Open data: It was unclear to me how I could reproduce or replicate this analysis, and I was frustrated to see a "data available on request" statement when this sort of statement violates the Copernicus data policy. While I appreciate the data is available on a tape drive somewhere in DKRZ, no instructions are given as to where to find it, and no DOI or similar is given. The authors should upload the model output to a public and persistent repository such as Zenodo, where it can be reversioned as the review process progresses. If a request is indeed necessary to publish this data on Swift, this is that request! I also note that no code was presented for review, which further hinders my ability to replicate and validate the analysis.*

[REPLY] Albeit the model data were solely produced for the current sensitivity study and will very likely not be used beyond that, neither by us nor by anybody else, we provide now data citations with persistent URLs for all model data. Information on how to access the model code has been added in the revised version of the manuscript.

*Ice types: the MOSAiC floe was located on second-year ice, and this should be mentioned since it results in a considerably less saline snowpack which will affect the temperature dependence of the albedo parameterization. I could not see in the manuscript what the ice type was for the Svalbard flights which led to the development of the new parameterization, but this should be stated and compared to the MOSAiC floe. If all of this takes place of second year ice, how relevant is it to first-year ice which is increasingly dominant in the Arctic?*

[REPLY] The new snow albedo parameterization was derived by Jäkel et al. (2019, https://doi.org/10.5194/tc-13-1695-2019) from measurements during the ACLOUD/PASCAL campaigns. The primary ice type during these campaigns was mostly first-year ice of varying thickness (Nicolaus, 2018, https://doi.org/10.1594/PANGAEA.889264). The results of the present study show that the new snow albedo parameterization also leads to improvements over second-year ice. It should be noted that there is no distinction of ice types in the model. The sea ice in a model grid cell usually represents a conglomerate of first-year and multi-year ice with different surface characteristics. The parameterization needs to reproduce the mean effect of

all these characteristics. Different temperature dependence of the albedo of different surfaces is certainly relevant for the overall performance of the albedo parameterization, but not relevant for the present study which emphasizes the importance of the cloud dependence.

*Figure 3: it seems like, if this pattern continues, the difference in SIV might continue to increase? Hopefully it will level off to some stable difference value, which will more realistically represent the impact of this parameterization. By simulating such a short period, it seems like you're simulating what might be referred to as a "transient response" of the system to your changes, rather than an "equilibrium response". And I think it's the equilibrium response that we should be really interested in? This should be discussed.*

[REPLY] It is right that the new albedo parameterization ends up in a new equilibrium state with thinner sea ice and less SIV. We demonstrate this in the revised version of the manuscript using the aforementioned new long-term sensitivity simulations (new Figures 4 and 5).

*Internal variability: I'm not an expert on regional climate models, but my understanding is that they do show some internal variability. I.e. if you ran the model with slightly different initial conditions or boundary forcing, you would get a different figure for the impact of this albedo parameterization that is larger than the scale of the perturbation to the conditions/forcings? Reading Rinke et al. (2004 Clim Res; and noting that Dr. Rinke is on this paper) it seems like this sensitivity of the findings to small differences and uncertainty in the initial/boundary conditions should at least be mentioned.*

[REPLY] Since the two simulations use identical initial conditions and identical boundary forcing, the differences between the two simulations are solely a response to the changed albedo parameterization. We have emphasized this fact in the revised version of the manuscript. The general role of the initial conditions in coupled regional climate models was investigated by Dorn et al. (2007, https://doi.org/10.1029/2006JD007814) and the role of internal model variability in the simulation of Arctic sea-ice extent and volume was demonstrated by Dorn et al. (2012, https://doi.org/10.5194/tc-6-985-2012). We refer to these papers when discussing the new long-term sensitivity simulations. The new long-term sensitivity simulations show clear changes in sea-ice volume and extent due to the revised snow albedo parameterization. These changes are much larger than the mean ensemble standard deviation of a 10-member ensemble of simulations with different initial conditions, indicating that the initial conditions are of no importance after sufficient spin-up time ($\sim 10$ years). In order to quantify the statistical significance of changes in sea-ice volume and extent due to the revised snow albedo parameterization with respect to interannual variability, we determined the 95 % and 99 % confidence intervals of the differences between the two long-term sensitivity simulations from the time series of the respective monthly means according to Welch's t-test. The results are shown in the new Figure 6 and substantiate that the sea-ice changes are statistically significant.

*This is of course clearly related to my previous concern about interannual variability (which could be viewed as an expression of internal variability). But I think it's worth me explicitly asking this question: for the year 2019/20, how sensitive are your SIE/SIV findings to small changes to the initial or boundary conditions? If they are significantly sensitive, then that would limit the power of this study.*

[REPLY] To demonstrate that the response of the sea ice to the changed albedo parameterization does not significantly depend on the sea-ice initialization, we have repeated the sensitivity

experiment using basically different initial data (ORAS5 data versus model data). The new pair of simulations is discussed in the revised version of the manuscript and indicates that the specific initial conditions may have impact on the magnitude of the sea-ice loss, but not on the sea-ice loss in general, since the temporal evolution is qualitatively similar.

*Sea ice initialisation: I couldn't find any information on how the sea ice was initialised in this experiment. Given this study is about the timing and nature of snowmelt onset, it seems obvious that the initial state of the sea ice in terms of its extent/thickness distribution and snow depth are relevant. The authors should specify how they initialised the sea ice cover, and what biases may be contained in this initialisation. Along these lines, the general evolution of the sea ice both in the model and reality for the study period should be briefly described.*

[REPLY] Information on the sea-ice initialization is given in the revised version of the manuscript for all simulations. The new pair of simulations with basically different initial conditions (see previous response) demonstrates that the choice of initial conditions has only minor impact on the results.

*I found the comparison of the model parametrization with MOSAiC data slightly odd. I agree with what you've written about the fact that the model timing won't agree with the MOSAiC data so a statistical comparison is necessary. But why didn't you just evaluate the quality of the albedo parameterization based on the observed variables in Table 1? Seems to me that would be the first step - test the parametrization by calculating the albedo with the in-situ observed snow temperature and cloud covers, then see if the model also gets it with its own modelled snow temp and cloud cover.*

[REPLY] The suggested first step, to test the parametrization by calculating the albedo with the in-situ observed snow temperature and cloud cover, was already done by Jäkel et al. (2019, https://doi.org/10.5194/tc-13-1695-2019) in the course of the parameter adjustment and was recently also applied to the complete albedo parameterization and to other data sets from different years and seasons by Jäkel et al. (2024, https://doi.org/10.5194/tc-18-1185-2024). We refer to these papers in the revised version of the manuscript. In the present paper, we take the second step, evaluating the albedo parameterization in fully coupled model simulations, and above all, we demonstrate how important it is (or could be in other models) to consider a cloud dependence in the snow albedo parameterization of a coupled model system. The latter has not yet been demonstrated and is the main purpose of the manuscript.

**Reply to Referee #2**

We thank Referee #2 for the time and effort he/she spent reading the manuscript and providing some major comments. Referee #2 noted a few times that there will be a second round with specific comments, but it remains unclear what he/she actually expects. Specific suggestions as to how we can or should improve the manuscript are missing. The following point-by-point response to the referee's comments (in italics) is therefore more what we actually did to overcome the referee's concerns about the manuscript. Since the changes/additions made in the manuscript are considerable, we can not mention all of them in detail and refer the reader to the marked-up manuscript version provided together with this response.

**Major Comments**

*Although the paper shows promise towards better understanding of how temprature affects albedo, I find the analysis incomplete especially with how the analysis is generalized just from one year of data from MOSAiC.*

[REPLY] The temperature effect on the albedo is not the topic of the present study. The study only concentrates on the cloud dependence of the snow albedo. Although the albedo values for cold, dry snow and warm, wet snow were modified, they were basically only separately defined for overcast and non-overcast conditions. When averaging the albedo for overcast and non-overcast conditions relative to the frequency of the conditions, the albedo values are almost equal in the two parameterizations. Changes in the temperature dependence take only effect in conjunction with the cloud dependence. All differences between the model simulations can thus be considered as a consequence of the additionally introduced cloud dependence. We have tried to make this point clearer in the revised version of the manuscript. We have also realized the need to explain the purpose of the paper in more detail. One purpose is to demonstrate that the consideration of a simple cloud-cover dependence in the broadband snow albedo parameterization is able to emulate the cloud effect on snow surface albedo reasonably. The main purpose, however, is to demonstrate that this rather minor modification of the snow albedo parameterization has a significant impact on the sea ice in a coupled model system due to its influence on the positive surface albedo feedback. To avoid misunderstanding, we state this purpose explicitly in the revised version of the manuscript. Furthermore, the analysis of the effects of a cloud-dependent snow albedo parameterization on the sea ice has been expanded in the revised version of the manuscript by two long-term sensitivity simulations for the period 1979-2021, one with the original snow albedo parameterization, the other with the revised snow albedo parameterization. The results of the two simulations are discussed in the revised version of the manuscript and confirm our assumption that the amplified surface albedo feedback due to the cloud-cover dependence of the snow albedo parameterization results in a new quasi-stationary equilibrium in sea ice with mostly thinner ice. This is now demonstrated by determining the statistical significance of the differences in SIV and SIE between the two long-term sensitivity simulations (shown in the new Figure 6).

*Also, with the amount of meteorological parameters collected during MOSAiC, why is only temperature and clouds, the only two parameters that are being investigated? Aren't there other met and geophysical parameters that affect albedo?*

[REPLY] As aforementioned, the study only concentrates on the cloud dependence of the snow albedo. Temperature effects are present in both parameterizations and can not be evaluated separately without further sensitivity experiments in which the temperature dependence of the albedo is independent from the cloud dependence. There are certainly other parameters that affect the snow albedo, as for instance black carbon deposits, the solar zenith angle, or snow age and depth, but such dependences are not included in the present snow albedo parameterization. Therefore, there is little point in evaluating the albedo against corresponding MOSAiC data.

*One of my concerns is that, the paper idealizes data from MOSAiC as representative at pan-Arctic scales, but I feel that a more exhaustive analysis from different Arctic sectors need to be incorporated to generalize your findings at a pan-Arctic scale.*

[REPLY] The new parameterization was derived from observations north of Svalbard in May/June 2017 by Jäkel et al. (2019, https://doi.org/10.5194/tc-13-1695-2019) and is in the present paper evaluated against independent data obtained from observations in the central Arctic in 2020. This is the regular procedure, detecting a physical principle from one data set and evaluating it against an independent other data set. Unfortunately, there are not many observations from the central Arctic against which the new parameterization can be evaluated. A more exhaustive evaluation of the complete albedo parameterization was recently made by Jäkel et al. (2024, https://doi.org/10.5194/tc-18-1185-2024). They evaluated the parameterization in offline and online application also against other Arctic data sets from different years and seasons. We refer to this paper in the revised version of the manuscript. Beyond that, the generalization of our findings is valid as long as there is no reason that indicates the contrary. The cloud dependence of the surface albedo is based on physical principles and was already demonstrated in several earlier studies that didn't make use of MOSAiC data. The physical principles are briefly explained in the introduction of our paper and a number of the earlier studies are cited (e.g., Grenfell and Perovich, 2008; Gardner and Sharp, 2010; Stapf et al., 2020). The theory that the cloud dependence of the snow albedo holds true for the entire Arctic is therefore not unlikely. Nevertheless, we have tried to improve our manuscript by revision of the introduction, by adding a detailed description of the sea-ice albedo parameterization in an own section, by adding and discussing the new long-term sensitivity simulations, by more in-depth conclusions, and by adding further references where appropriate. We hope that the revised version of the manuscript better supports our theory that the consideration of the cloud dependence of the snow albedo is important for coupled model simulations.

---

## Author Response (AR2)

**Reply to the Comments from Referee #1 (Report #2)**

We thank Referee #1 for the time and effort he/she spent reading the manuscript again and providing further suggestions for improvements. Please find below a point-by-point response to the referee's comments which are shown in italics.

**SIE vs SIA**

*Presumably you've calculated SIE as the total area of all grid cells with at least 15% SIC. However, "sea ice extent is a strongly grid-dependent, nonlinear quantity" (Notz et al., 2020). That paper also points out that the observational spread for trends in SIA is smaller than it is for trends in SIE, so strongly advocates the use of SIA over SIE in this sort of investigation. I think it's critical to state whether ORAS5 (your comparison data set) has the same grid as your HN model output, because if it doesn't then your HIRHAM-NAOSIM SIE values are not comparable to the ORAS5 ones (even though they may be close). Just looking at Zuo et al, it looks like the ECV data coming out of ORAS5 might be given on a 1x1 degree grid whereas yours is much finer? In light of that, and particularly as your grid cells are non-uniform in size, it would be much better for this paper to present its results in terms of SIA. I don't think it will change your results much (although it might reconcile the HIRHAM/NAOSIM trends to ORAS5 a bit?. But it will make the results more rigorous and adhere to SIMIP best practice in terms of model output intercomparison. This change should be made prior to publication.*

[REPLY] We have followed the referee's suggestion and have replaced SIE by SIA throughout the manuscript. In addition, we now mention in the figure captions that "ORAS5 data were bi-linearly interpolated to the NAOSIM grid using distance-weighted averages for missing values".

**Subjective Language**

*The word "significant" appears eleven times in this manuscript. It's used twice as part of the phrase "statistically significant" (which is of course fine), but in the other nine times I think it's used quite subjectively and should be reconsidered. Many journals now (in my view rightly) don't allow this. For example, Nature journals now do not allow "significant" to be used without an accompanying p-value. I've already mentioned that describing your changes as a "major improvement" also seems subjective - I think it's safe to say things have improved, but whether your improvements are major vs minor is perhaps in the eye of the beholder. The same with the description of the "simple, but innovative cloud-cover dependence". I agree it's simple, but whether it's really that innovative or not is probably something for the cover-letter rather than peer-reviewed literature (so I suggest you remove that word).*

[REPLY] We have removed words like "significant" or "significantly" when we are not explicitly referring to a statistical test (three times). In the other case, we have added the attribute "statistically" in order to clearly indicate that the significance was statistically tested (seven times). Furthermore, we have removed the attributes "major" for the improvement and "innovative" for the cloud-cover dependence.

**ORAS5 biases**

*You should discuss some implications of using ORAS5 as a benchmark from which to calculate bias. Per your rebuttal, you're implicitly treating it as "the truth" for purposes of measuring whether HNnew is better than HNold. I think doing that is fine, but it needs context.*

[REPLY] We have expanded the introduction of ORAS5 as reference data set for the evaluation of sea-ice volume and sea-ice area (lines 199–208 in the revised version). We refer now to biases in ORAS5 as discussed by Tietsche et al. (2018) and Zuo et al. (2019), and we explicitly point to differences in sea-ice volume between ORAS5 and PIOMAS.

*For example regarding Figure 6, there is a report which claims ORAS5 is too thick in winter. How do biases in the product affect your findings regarding the model? What's the risk that the albedo params are being tuned to match a dataset that is itself biased?*

[REPLY] Indeed, ORAS5 tends to overestimate the thickness of sea ice as indicated by Tietsche et al. (2018); on the other hand, PIOMAS rather tends to underestimate the ice thickness as indicated by Schweiger et al. (2011). Assuming that we had used PIOMAS as benchmark, the bias reduction in HNnew compared to HNold would have been even more obvious, considering that HNold shows even thicker sea ice than ORAS5. This means that there is no risk when we claim that HNnew performs better than HNold.

*Could you also elaborate on what you mean on line 211? I understand that model biases are larger than the spread of observational products, but I don't understand what you mean about "qualitatively equal model biases would appear" if you evaluated against the observational products.*

[REPLY] This statement has been completely removed from the manuscript.

**Initial Conditions**

*I'd still like to know more about the "initial conditions" i.e. is the snow realistically deep, is the ice in a realistic place and does it have realistic thickness. From L100 I gather that P1 was initialised from a long-term run of NAOSIM but the others from ORAS5. For P1 I'm less concerned about the initial conditions because of the spinup time. But I think it is relevant for P2 & P3: was the initial snow depth distribution also taken from ORAS5? For P2 & P3 it would be good to see a supplemental two-panel figure with a map of (a) initial SIT distribution (b) initial snow depth distribution. I imagine the SIT distribution is sufficiently realistic, but I do think it's important when experimenting on such a short timescale to know about the initial snow depth. I.e. how deep is it and where is it. Because that may affect the timing of snow melt onset, which is one subject of the paper.*

[REPLY] In the first round of the peer review process, it was requested to demonstrate that the findings do not depend on the specific initial conditions. Therefore, an alternative method of initializing the model was chosen for P2, namely using (more realistic) ORAS5 fields instead of (more consistent) model restart fields as in P1. In contrast to P1, P2 is initialized with zero snow thickness. We explicitly mention this fact now in line 92.

As snow rarely survives the warm season, the snow thickness distribution in all simulations, be it from P1 or from P2, is similar after the first melting season due to the applied nudging to ERA5, which leads to similar snow fall patterns in the model. Whether the simulated snow distribution is realistic or unrealistic is an open issue. Ongoing studies show that the evaluation of the modeled snow fall or snow distribution represents a fundamental problem, not only from the model side but also from the observational side. Tackling this problem goes far beyond the scope of this manuscript.

In contrast to snow, sea ice possesses some kind of memory effect, namely in terms of multi-year sea ice. The initial ice thickness distribution is therefore not irrelevant for a few years. The fact is, the initial ice thickness distribution is completely different in P1 and P2, and nonetheless, the effect of the revised snow albedo parameterization in both pairs is comparable. We have added maps of the initial ice thickness distribution in P1 and P2 as a supplemental figure to the manuscript. Showing the initial snow thickness distribution is redundant due to the afore-mentioned reasons, and showing any initial conditons from P3 is redundant as well, because the 10-year-long spinup time is long enough to forget any initial state.

**Other**

*L218: it would be good to get some numbers in here for slopes. At what rates are HNnew, HNold & ORAS5 declining?*

[REPLY] Corresponding numbers have been added (lines 214–215 in the revised version).

*L220: I guess this is because of the nudging scheme? I.e. the place where sea ice is, and is not, is driven by atmospheric circulation. But the thickness is driven by the radiative balance.*

[REPLY] Only P1 and P2 were carried out with nudging, because these simulations were primarily intended for the comparsion with MOSAiC measurements. The P3 simulations (which are discussed in this sections) were running without nudging, meaning that the atmospheric circulation not only drives the sea ice, but also responds to the sea-ice conditions. Nevertheless, we agree that the ice area is more strongly controlled by the atmospheric circulation than the ice volume.